# Evaluating Tumour Mutational Burden as a Key Biomarker in Personalized Cancer Immunotherapy: A Pan-Cancer Systematic Review

**DOI:** 10.3390/cancers17030480

**Published:** 2025-02-01

**Authors:** Anca Zgura, Stefania Chipuc, Nicolae Bacalbasa, Bogdan Haineala, Anghel Rodica, Vâlcea Sebastian

**Affiliations:** 1Department of Oncology-Radiotherapy, Prof. Dr. Alexandru Trestioreanu Institute of Oncology, “Carol Davila” University of Medicine and Pharmacy, 020021 Bucharest, Romania; anca.zgura@umfcd.ro (A.Z.);; 2Prof. Dr. Alexandru Trestioreanu Institute of Oncology, 022328 Bucharest, Romania; 3Department of Surgery, “Carol Davila” University of Medicine and Pharmacy, 050474 Bucharest, Romania; nicolae.bacalbasa@umfcd.ro (N.B.); sebastian.valcea@gmail.com (V.S.); 4Department of Urology, “Fundeni” Clinical Institute, “Carol Davila” University of Medicine and Pharmacy, 050474 Bucharest, Romania; bhaineala@gmail.com

**Keywords:** tumour mutational burden (TMB), immune checkpoint inhibitors (ICIs), systematic review, biomarkers, cancer immunotherapy, personalized medicine

## Abstract

Cancer treatment using immune checkpoint inhibitors (ICIs) has shown great success, but not all patients respond equally. Tumour mutational burden (TMB) is a measure of genetic mutations in cancer cells and is used to predict whether ICIs will be effective. While TMB is a reliable predictor in lung cancer and melanoma, its usefulness in breast and prostate cancers remains uncertain. This study reviewed existing research to assess how well TMB pre-dicts treatment outcomes in these four cancer types. The results confirmed that high TMB is linked to better re-sponses in lung cancer and melanoma. However, findings for breast and prostate cancers were inconsistent, sug-gesting that other factors may influence treatment success. Differences in how TMB is measured also make com-parisons difficult. To improve cancer treatment, future research should focus on refining TMB assessment methods and identifying additional markers that can help personalize therapy.

## 1. Introduction

As cancer immunotherapy continues to revolutionize treatment paradigms, TMB has gained prominence as a potential biomarker for predicting the efficacy of ICIs. Extensive research has established TMB as a reliable predictor of responses in lung cancer and melanoma, where higher mutational loads correlate with better clinical outcomes. However, its applicability to other cancers, such as breast and prostate, remains ambiguous, due to inconsistent findings and methodological disparities. This uncertainty underscores the need to evaluate the broader relevance of TMB as a predictive marker, especially in cancers where its utility is unclear. By systematically reviewing the existing studies, this work aims to bridge the gaps in knowledge, refine the understanding of TMB’s role in diverse cancer types, and propose new avenues for integrating it with other biomarkers. A comprehensive synthesis of evidence is crucial for advancing personalized immunotherapy and addressing variability in treatment outcomes.

Objectives: To evaluate the predictive value of tumour mutational burden for immune checkpoint inhibitor therapy across four major cancer types: lung, melanoma, breast, and prostate. To identify the patterns, inconsistencies, and limitations in existing studies on TMB as a biomarker. To propose directions for future research to improve the integration of TMB with additional biomarkers and refine its use in personalized cancer therapy.

Key questions:What is the evidence supporting TMB as a predictive biomarker for ICI therapy?How does the predictive utility of TMB differ between cancer types, and what factors might explain this variability?How can TMB be integrated with other biomarkers to improve its predictive accuracy for breast and prostate cancers?

## 2. Methodology

### 2.1. Literature Search and Selection

To evaluate the role of Tumour Mutational Burden (TMB) in predicting responses to immune checkpoint inhibitors (ICIs) in four major cancer types (non-small-cell lung cancer, melanoma, breast cancer, and prostate cancer), we conducted a systematic review of the scientific literature. Key databases, including PubMed, Scopus, and Web of Science, were searched for articles published between 2010 and 2023, using keywords such as “Tumour Mutational Burden”, “immune checkpoint inhibitors”, “NSCLC”, “melanoma”, “breast cancer”, “prostate cancer”, and “biomarkers”. This review was performed in accordance with the PRISMA (Preferred Reporting Items for Systematic Reviews and Meta-Analyses) guidelines and has not been registered (Figure 1).

### 2.2. Inclusion and Exclusion Criteria

We included studies published in English, which were focused on the relationship between TMB and the response to ICIs in at least one of the four analyzed cancer types.

Inclusion criteria:Published in English;Focused on the relationship between TMB and responses to ICIs in at least one of the four analyzed cancer types;Used standardized methods for measuring TMB (whole exome sequencing or approved gene panels);Reported clinical data on the efficacy of immunotherapy based on TMB levels.

Exclusion criteria:Did not include clinical data related to patients treated with immunotherapy;Were narrative reviews or commentary articles without empirical data;Did not use validated methods for TMB assessment.

### 2.3. Data Extraction and Analysis

The data from the selected studies were extracted and organized by cancer type and reported TMB levels (low, intermediate, or high). Key outcomes included response rates to immunotherapy, progression-free survival (PFS), and overall survival (OS), correlated with TMB levels. The methods used to assess TMB, such as whole exome sequencing (WES) and gene panel tests (e.g., FoundationOne CDx), were also compared.

### 2.4. Quality Assessment of Studies

The quality of the included studies was assessed using criteria such as sample size, methods for TMB assessment, and the robustness of statistical analyses. Studies were rated according to the Newcastle–Ottawa Scale, and studies with lower scores were excluded from the final analysis to ensure scientific rigour. All sources were manually screened by the authors, independently.

### 2.5. Statistical Analysis

To evaluate the association between TMB and clinical outcomes, Cox regression models were used to calculate hazard ratios (HRs) for survival and response to immunotherapy. Meta-analyses were conducted using Review Manager (RevMan 5.4), and heterogeneity across studies was assessed using the I^2^ statistic. A significance level of *p* < 0.05 was considered statistically significant.

Following the individual analysis conducted by each author, 44 articles were considered suitable for inclusion in this meta-analysis. For the presentation of results, several key concepts were used, such as “TMB efficiency” and “role of immunotherapy”, as well as statistical data, such as OS and PFS. The necessary data extracted from the original articles for the meta-analysis were used based on the citation in the text. Additionally, no original figures were used, and the meta-analysis contains a single table comparing three major studies focused on lung cancer.

#### 2.5.1. What Is TMB?

TMB can be defined as a tumour’s total number of mutations. The TMB value can vary depending on the technique used to measure it [1]. Currently, the measurement is performed using NSG (next generation sequencing). Several NGS approaches are developed, and the target region ranges from genome-wide analysis (WGS—whole genome sequencing) to whole exome analysis (WES—whole exome sequencing).

Evaluating TMB is crucial in personalizing oncological treatments, particularly immunotherapies. NGS methods are fundamental for determining the number of somatic mutations in tumour DNA. These methods differ significantly depending on the targeted genomic region:Whole Genome Sequencing (WGS) provides a comprehensive analysis, including both coding and non-coding regions of the genome;Whole Exome Sequencing (WES) focuses on coding regions, which represent only 1–2% of the genome, but contain most cancer-related mutations.

Recently, the FDA approved FoundationOne CDx and MSK-IMPACT [2,3], 2 gene panel assays designed to identify genetic alterations in solid tumours. These assays are used in clinical settings due to their precision and ability to streamline TMB assessment. However, the diversity of platforms and methodologies used in different studies has led to substantial variability in TMB results, complicating the comparison of findings and their clinical applicability.

This highlights the pressing need for standardized protocols at an international level. Standardization would ensure consistency in the methods used, including sequencing platforms, bioinformatics pipelines, and TMB cut-offs. Collaborative efforts between regulatory agencies, academic institutions, and industry stakeholders are essential to establish these standardized frameworks. Furthermore, the integration of validated assays like FoundationOne CDx into unified protocols could serve as a model for harmonizing TMB evaluation in clinical and research settings.

#### 2.5.2. Testing Method: WES

Whole exome sequencing allows for the exploration of all the protein-coding regions of the human genome. This technology facilitates the examination of genetic mutations associated with cancer, abnormalities that are mainly located in the exome regions. In WES, the focus is on specific regions of the genome, the protein-coding fragment. This allows us to identify genetic abnormalities that will impact protein function [4]. WES is a powerful tool for the detection of various genomic changes, both in coding and non-coding DNA that are influential in cancer development [5]. Changes in the exome can lead to different amino acid substitutions in protein. This event can lead to the weakened activity of multiple tumour suppressors, such as APC in colorectal cancer, VHL in renal cell carcinoma, or BRCA in breast cancer [6,7,8]. There are also modifications in cell cycle regulators, such as TP53 or RB1, and in repair mechanisms, which will predispose the patients (them) with those mutations to cancer development. By measuring the TMB (mutational burden), we can monitor the activity of those systems [9,10].

Currently, there are 2 primary types of NGS methods: DNA amplification-based sequencing (Illumina, Ion Torrent) and single molecule real-time sequencing (Pacific Biosciences, Oxford Nanopore). The tissue samples analyzed can be liquid-based (blood sample), freshly frozen, formalin-fixed or paraffin-embedded (FFPE). Each type of sample requires its own specific isolation kits [4].

The first step of WES is an adequate examination of the sample, conducted by a pathologist. The used sample should contain a proper amount of tumour cells to differentiate between germline and somatic mutations. It is important to keep in mind that the DNA quality deteriorates with time and after FFPE conservation [4]. After the examination of the samples, the data processing starts, usually with a quality control. The low-quality reads are eliminated. The next step is to align the reads with a reference genome, then a second quality control, and finally, the removal of the duplicated reads.

The main advantage of the WES technique is that it can scan the entire genome of a sample and provide information about the low-frequency mutations, which can collectively determine a phenotypic appearance [11].

#### 2.5.3. Correlation Between TMB and Immunotherapy

Nowadays, immune checkpoint inhibitors have become the standard therapy for various solid tumours, such as melanoma, NSCLC, or renal cell carcinoma [12]. The response to this therapy can be measured by a reliable biomarker, TMB, which represents the total number of mutations per coding sequence in the tumour genome. Currently, using WES to detect TMB is a widely accepted method [4]. Despite the potential of TMB as a predictive biomarker, a lack of consensus remains regarding its definition, determination method, and adequate cut-off values. For example, Foundation Medicine divided the TMB into 3 categories: high TMB (>20 mut/Mb), intermediate TMB (6–19 mut/Mb), and low TMB (less then 5 mut/Mb) [13]. Regarding this issue, the Friends of Cancer Research have established a team whose primary goal is to standardize the use of TMB [14].

#### 2.5.4. The Interpretation and Reporting of the TMB Value

TMB quantification is influenced by 4 main factors [1]:Tumour purity: This represents the overall percentage of cancerous cells within a tumour sample. This measurement is analyst-dependent and can lead to errors since the used sample may not represent the tumour’s region that will be analyzed;Library construction and sequencing: This is represented by DNA fragments with a defined length that will be analyzed using various bioinformatics programmes;The pipeline used to call mutations: This represents the algorithm used to remove germline variants. This is a vital step in the identification of different somatic mutations that are responsible for producing tumour neo-antigens. These antigens will be eventually recognized as non-self by the immune system;The capacity to extrapolate TMB values from the restricted genomic space sampled by gene panels: This step is based on the in silico analysis performed on samples to determine the concordance between WE-based TMB and panel-based TMB [1].

#### 2.5.5. Can We Use TMB as a Predictive Biomarker?

Typically, T cells recognize the different neo-antigens produced by various mutational mechanisms and presented by MHC molecules from the cancerous cell’s surface, and target those cells for destruction. To evade T cells and to supress the immune system, a tumour has the capacity to produce proteins that, normally, function as checkpoints that attenuate immune responsiveness. The main reason for using immune checkpoints is to block the interaction between T cells and tumour proteins, with the aim of reactivating the immune system. Once the immune system is reactivated, T cells can differentiate normal cells from cancerous ones. This process is facilitated by the presence of immunogenic antigens on the cell’s surface. Since these molecules arise from mutations, the more neo-antigens that are present, the higher the TMB. This is the hypothesis that supports the idea that the higher the TMB, the greater the chances of responding to a treatment based on ICIs [1]. However, there is also evidence that approximately 60% of patients with high-TMB do not have a malignancy that responds to ICIs [15].

To determine whether TMB can be used as a predictive biomarker or not, several studies were conducted. The most informative study is that conducted by Hao-Xiang Wu and his team [16]. In this study, 20 primary solid cancers from 6035 patients were analyzed, and for each type, the impact of TMB on the overall survival (OS) was evaluated using the Kaplan–Meier method. Survival analysis showed in the end that TMB has a significant impact on OS in 14 cases out of 20. According to this, the impact of the TMB was classified into 3 categories: the TMB-worse group (it includes 8 types of malignancies—the patients with a high TMB have a poorer prognosis compared to those with a lower value), the TMB-better group (6 types of cancer—the patients with a higher TMB have a better prognosis and a decreased mortality rate), and the TMB-similar group, where the value of TMB did not have an impact on OS [16].

## 3. Lung Cancer and TMB

Pulmonary cancer is the most common cause of death from cancer worldwide [17]. It includes different histological subtypes, of which non-small-cell lung cancer (for short NSCLC) represents approximately 85% [18]. Only a small percentage (20–25%) of patients with NSCLC have an early-stage diagnosis. In this case, elective resection is performed with curative intent [17]. Over recent years, a better understanding of NSCLC’s biology led to the identification of various predictive biomarkers, for example, EGFR, BRAF mutations, or ALK and ROS1 rearrangements. Due to these findings, several target therapies have arisen. In NSCLS, a targetable alteration is found in approximately 50% of patients, so for the other half, it is imperative to find a biomarker to improve the clinical outcomes [19].

Nowadays, the only biomarker approved is PD-L1 expression, assessed by immunohistochemistry (IHC), an inexpensive technique that is performed using standard histopathology equipment. PD-L1 expression is an important criterion during the process of splitting patients between those who will receive treatment with Pembrolizumab in monotherapy (in this case the PD-L1 cut-off is ≥50%) [20] and those who will receive a dual immunotherapy Nivolumab plus Ipilimumab (PD-L1 cut-off of 1%) [20]. The cut-off values are explained by the multitude of assays used by investigators in clinical trials. The variable need to be standardise in daily practice (PD-L1 needs standard testing in clinical practice). For this reason, there is a need for identifications of other biomarkers, including TMB [19].

Given the variety of treatment options available, biomarkers are essential to determine which patient subgroup is most likely to benefit from a particular therapy. Although not ideal, high PD-L1 expression levels can help select patients who are likely to benefit from monotherapy with pembrolizumab. TMB is a promising biomarker, but assays and criteria for defining high TMB need standardization, and obtaining sufficient tissue samples can be challenging.

In the current literature, there are three main studies that suggest the possibility of TMB utilization as a biomarker in NSCLC.

All three studies (Table 1) included a large number of patients, evenly distributed between the study arms. They presented similar inclusion and exclusion criteria. The results of these studies concluded that immunotherapy is a better therapeutic option than chemotherapy in NSCLC, and that dual immunotherapy is superior to monotherapy compared to chemotherapy (median PFS 4.2 m in Nivolumab group vs. 5.9 m in chemotherapy; median OS 14.4 m for Nivolumab vs. 13.2 m for chemotherapy) [21]. Additionally, the use of TMB as a potential marker for predicting treatment response was also discussed in these studies. The TMB cut-off was universally accepted to be 10 mutations/Mb, with higher values not being associated with improved outcomes [22]. For the Nivolumab + Ipilimumab combination, the 1-year PFS was longer than in the chemotherapy arm (42.6% vs. 13.2%) and the median PFS was 7.2 m for those patients with TMB ≥ 10 mut/Mb. For those with less than 10 mut/Mb, the median PFS was 3.2 m in the Nivolumab + Ipilimumab arm [23]. CheckMate 568 [22] had an interesting finding regarding the biomarkers. The medical team observed that, regardless of PD-L1’s value, ORR, as a secondary endpoint, was higher in patients with an increased TMB (≥10 mut/Mb). As a primary endpoint, PD-L1 ≥ 1% had a better outcome, with an ORR 41% vs. 15% (<1%). Based on these results, we can conclude that, in addition to PD-L1, we can use TMB to guide our treatment choices and to predict our patients’ outcomes.

Despite these positive results, there are also studies that suggest the opposite. These inconsistencies highlight the importance of large-scale analyses to universally establish the predictive value of TMB as a predictive biomarker.

To support the idea of a negative correlation, we will present the conclusions of a retrospective study performed on 136 NSCLC patients [24]. Chae et al. undertook ctDNA testing in one institution, with an additional validation cohort analyzed in another institution. The ctDNA TMB was measured by counting all detected mutations of the sequencing length. After the analysis, the conclusion was that a higher ctDNA TMB was significantly linked with a history of smoking (*p* < 0.05, chi-squared test). Among patients treated with immune checkpoint inhibitors (*n* = 20), increased TMB was correlated with shorter progression-free survival (PFS) and overall survival (OS; 45 vs. 355 days; hazard ratio [HR], 5.6; 95% confidence interval [CI], 1.3–24.6; *p* < 0.01, and OS 106 days vs. not reached; HR, 6.0; 95% CI, 1.3–27.1; *p* < 0.01, respectively). There was also a small number of patients (*n* = 12) among whom there was non-significant correlation. In conclusion, this study suggested that a higher TMB can be associated with a negative clinical outcome. Additionally, the authors of the study emphasize the importance of large-scale studies, given that their analysis had several limitations, such as the insufficient length of the tested DNA and the use of a single type of commercially available kit for this testing [24].

## 4. Melanoma and TMB

Apparently and encouragingly, TMB can be used as a predictive biomarker for ICIs and is likely to be included in future treatment protocols for various types of malignancies. Based on the genetics of melanoma, we will be able to predict the response to treatment (based on CTLA-4 blockade). However, the association between high TMB (in melanoma) is not the main factor for predicting a better response on ICIs therapy. Further research is needed to better understand the influence of the somatic neoepitopes shared by patients with a prolonged benefit from ICIs [25]. For understanding the role of TMB, Dousset et al. [26] included 102 patients with advanced malignant melanoma in their study. Their clinicopathologic characteristics, as well as tumour genomic outcomes, were collected. An important aspect of this study was the patients’ distribution based on sun-exposure pattern in three different groups: chronic exposure (head and neck), intermittent exposure (arms, chest/torax/trunk, and legs) and protected regions (feet, toes, soles of feet, genitals, and the mucosal and uveal regions). For these patients, TMB was analyzed on the recently metastatic sample prior to administration of PD-L1 inhibitor therapy.

The analyzed group was primarily composed of men (57%) and the median age at diagnosis was 59.3 years. The majority of melanomas were superficial (36%), followed by unknown primary site (19.6%) and nodular-type melanomas (16.8). The rarest locations were identified as follows: acral lentiginous (12 cases out of 102, 11.8%), mucosal melanoma (9 cases, 8.8%), uveal melanoma (5 cases, 5 cases of uveal melanoma represent 5% off all melanoma cases) and naevocytoid and desmoplastic, 1 case for each type. The BRAF mutation was evaluated as well, and a positive result was received in 34 cases. Out of 102 patients, 80 were treated with anti-PD-L1 monotherapy. TMB was assessed in 94 cases and the median value was 12.4 mut/Mb.

The primary endpoint of this study was to determine if the sun exposure influenced the TMB and consequently the ICI treatment response. The TMB was significantly higher in chronically exposed areas (37.2 mut/Mb vs. 13.6 mut/Mb and 4 mut/Mb, respectively). These results were according to the initial hypothesis; TMB is linked to sun exposure and UV signature. It was stated that, in cutaneous melanoma, TMB is higher than in other cutaneous, nonmelanoma tumours, due to the mutagenic effects of UV exposure. Both the UV signature of the primary site and the metastatic ones are associated with an increased TMB, suggesting that melanoma metastases carry the exact UV signature present in the primary site of origin [26].

## 5. Breast Cancer and TMB

One of the most common malignancies seen in women is breast cancer (BC), and various biomarkers, such as ER, PR, and HER2, are currently used for therapeutic decision-making processes [27]. Although there are several treatments protocols for BC, almost 30% of patients will eventually develop advanced disease, which requires another treatment approach. The reduced efficacy of targeted therapies and the relatively poor prognosis of advanced BC patients have underscored the need to explore new treatment approaches, including immunotherapy [28,29]. Recent studies have shown that Pembrolizumab and Atezolizumab plus Nab-Paclitaxel offers promising clinical benefits for patients with advanced triple-negative BC. As an initial step, PD-L1 was proposed as a biomarker for evaluating immunotherapy efficacy, but there were some concerns about its accuracy, since PD-L1 testing has its own challenges (variability among assays or lack of standardization) [29].

In BC patients, TMB has not been well characterized, but we will further discuss the conclusions of the initial analysis regarding its role in BC. Mei et al. [27] analyzed 62 advanced BC cases between January 2014 and June 2018. The samples were tested using Foundation Medicine and TMB by FoundationOne CDx next generation sequencing (NGS). The TMB values were classified into three main groups: low (1–5 mut/Mb), intermediate (6–19 mut/Mb), and high (≥20 mut/Mb). The demographic features of the study included median age (53.8 years old, range 30–78) and the majority were metastatic cases (49, 79.0%). A total of 52 (83.9%) cases were invasive ductal carcinoma (IDC), 6 cases (9.7%) were invasive lobular carcinoma, 2 cases (3.2%) were metaplastic carcinoma, and another 2 cases were neuroendocrine carcinomas [27].

From a biomarker perspective, we note that 36 cases (58.1%) were ER positive, 38 cases (61.3%) were PR positive, 5 cases (8.1%) were HER2 positive, and 22 cases (35.5%) were triple negative. Among 62 cases, 3 of them (4.8%) had high TMB, 27 (43.6%) had intermediate TMB, and the majority (32 cases, 51.6%) had low TMB. Since cases in the high and intermediate groups were not so numerous, the team decided to combine these two and compare them with the low TMB group.

As a first conclusion of this study [27], the team have shown that, through the evidence of the associations between increased TILs (Tumor-Infiltrating Lymphocytes) and the group of cases with intermediate/high TMB compared with the association between TILs levels and the low TMB group (*p* = 0.0018), there were no other correlations between TMB and other clinicopathologic characteristics. There were several gene mutations evaluated in this study. The second most important conclusion was linked to the commonly seen genetic mutation. The most common mutation identified among the 62 cases was TP53 (59.7%), followed by PIK3CA (33.9%). Interestingly, out of the six BC cases with BRCA (1/2) mutations analyzed, five had intermediate or high TMB, while only one case exhibited low TMB (*p* = 0.0002). A total of 34 DNA damage repair (DDR) genes were included in the NGS panel of this study, and a total of 13 cases exhibited at least one DDR gene mutation, while the remaining 49 cases did not show any DDR gene mutations. Clinicopathologic features and TMB were compared between cases with and without DDR mutations. BCs with DDR mutations had a higher TMB compared to those without DDR mutations (12.08 vs. 6.57 average mutations; *p* = 0.043). There were no differences observed for other clinicopathologic characteristics and the two groups [27].

Although this is one of the first studies on the impact of TMB in breast cancer, the authors sustain that more analysis should be carried out on this topic, given that immunotherapy is a relatively new option for BC treatment and that no major data are available in the literature. The limited sample size (*n* = 62) restricted the significance of this analysis. Future studies with larger cohorts are necessary to confirm these results.

## 6. Prostate Cancer and TMB

Prostate cancer is the second most common type of malignancy seen in males [30]. For this type of cancer, challenges regarding the treatment persist, particularly for castration-resistant prostate cancer (CRPC). Metastatic CRPC is associated with a poor prognosis, with a median survival period of less than 2 years [31]. Recently, immune checkpoint inhibitors targeting programmed cell death 1 and its ligand (PD-1/PD-L1) and Cytotoxic T-lymphocyte antigen-4 (CTLA-4) have shown promising preliminary results in various tumours. However, the effectiveness of immunotherapy remains limited by its low efficacy. Potential predictive biomarkers, such as tumour mutational burden (TMB), are evaluated here.

Graf et al. [32] carried out a study aimed at assessing the outcomes of patients treated with immune checkpoint inhibitors (ICIs) versus taxane chemotherapy, with a focus on tumour mutational burden (TMB)’s role. Immune checkpoint inhibitors (ICIs) can induce significant responses and provide long-term benefits in some patients with metastatic cancer who have undergone numerous prior treatments. However, the rate of clinical benefit varies significantly according to tumour type. Unfortunately, for patients with metastatic castration-resistant prostate cancer (mCRPC), the objective response rate to ICI treatments is reported to be around 3% for those without programmed cell death ligand 1 (PD-L1) expression and 5% for those with PD-L1-expressing tumours [33]. For this reason, there has been increasing interest in identifying other biomarkers that could pinpoint mCRPC patients who are more likely to achieve greater clinical benefits from ICIs compared to alternative treatments.

In this study, [32], a total of 741 men were evaluated between January 2011 and April 2021. They were subjected to genomic testing through the comprehensive genomic profiling (CGP) assays provided by Foundation Medicine. Patients were included in this study if they received either single-agent anti–PD-1 axis therapy or single-agent taxane in the mCRPC setting, and had their TMB assessed through tissue biopsy. The main clinicopathological characteristics were represented by a median age of 70 (ranges between 64 and 76 years) and the baseline median pretreatment PSA levels of 79.4 ng/mL. A total of 108 patients (18.8%) had ECOG scores of 2 or greater, and 644 patients (86.9%) had received prior systemic treatments for mCRPC. A total of 45 patients (6.1%) received ICIs, while 696 patients (93.9%) received taxanes. Patients who received ICIs and those who received taxanes showed no significant differences in age, pretherapy PSA levels, ECOG scores, prior NHT use, prior prescribed opioid use, or biopsy site. However, it is important to note that patients who received ICIs had higher TMB compared to those receiving taxanes (3.5 mut/Mb vs. 2.5 mut/Mb; *p* < 0.001).

PSA levels were evaluated in 607 patients. Among them, 14 patients had a TMB above 10 mut/Mb and were treated with ICIs. Of these, four exhibited a reduction in PSA levels by approximately 50%. In contrast, none of the patients with a TMB below 10 mut/Mb showed a reduction in PSA levels greater than 50%. For patients treated with taxanes, no relationship between the TMB value and PSA level was demonstrated [32].

The FDA-approved cut-off for TMB was 10 mut/Mb. Among its conclusions, the study showed that both TTNT (time to next treatment) and OS (overall survival) were adjusted according to received drug class (ICIs vs. taxanes). The patient group with TMB < 10 mut/Mb and ICI therapy had a worse average TTNT than those treated with taxanes (2.4 m vs. 4.1 m). The reverse pattern for TTNT was observed for those with TMB > 10 mut/Mb (8.0 m vs. 2.4 m). The OS evaluation did not present major differences between those with TMB < 10 mut/Mb, despite the treatment choice (median OS 4.2 m vs. 6.0 m). However, in those cases with TMB greater then 10 muts/Mb, the OS had an elevated value when ICIs were administered compared to taxanes (median OS 19.9 m vs. 4.2 m) [32].

Despite the evidence of improved clinical outcomes for the group with high TMB and ICI therapy, the study has some limitations. It is not a randomized study; the treatment was chosen by the clinician and the number of patients that received ICIs was reduced compared to the taxane group. Also, the biopsy timing was not considered. This leaves room for future studies to demonstrate the effectiveness of using TMB in a larger cohort of patients undergoing immunotherapy treatment.

## 7. Discussion

Representing some of the most severe forms of cancer globally, lung, breast, prostate cancer, and melanoma continue to pose challenges for clinicians when it comes to selecting the optimal management protocol. The key to a successful treatment lies in the individualization of the protocol, based on the clinical and pathological characteristics of each patient. From the perspective of evaluating treatment efficacy, there is an increasing need to identify a marker that can highlight the category of patients who are most likely to have a better therapeutic response.

Immunotherapy marks a significant advancement in cancer treatment. The mechanism is based on the reactivation of the tumour immune cycle, which will restore the body’s natural anti-tumour immune response. Currently, there are at least four types of immunotherapy strategies: immune checkpoint inhibitors (ICIs), such as programmed cell death protein 1 (PD-1) and Cytotoxic T-Lymphocyte Antigen 4 (CTLA-4), chimeric antigen receptor T cell therapy, tumour vaccines, and lastly, peripatetic immunotherapy. Although these therapies have significantly improved clinical oncology outcomes, not all patients have experienced the benefits. Therefore, it is essential to determine which patients are most likely to respond favourably to immunotherapy [34].

To date, PD-L1 is the only biomarker used to predict the response to ICIs. However, since its evaluation varies depending on the kit used by each medical team, there is a massive need for the detection of a reliable biomarker for predicting therapeutic responses. Lastly, several studies have detected another biomarker that can be used for predicting therapeutic outcomes. Tumour mutational burden (TMB) represents the total number of mutations (substitutions, insertions, or deletions) that occurs in a tumour sample [26]. TMB can be divided into three categories: low, intermediate, and high, depending on the cut-off value considered by each medical team. As observed in the section dedicated to NSCLC, the universally accepted cut-off value for TMB at this time is 10 mut/Mb, with higher values not being correlated with improved outcomes. This cut-off value itself represents a limitation of any study on this topic, as there is currently no consensus on this value.

Various studies have demonstrated that TMB offers some advantages over other biomarkers. Firstly, TMB can be measured in the blood, which is beneficial in cases where tumour tissue specimens are unavailable. Non-invasive blood-based TMB (bTMB) has emerged as a promising method for assessing TMB in clinical settings for immune checkpoint inhibitor (ICI) treatment. However, the reliability of bTMB measurements can vary due to several factors, including the technical aspects of the ctDNA assay, such as the sensitivity of variant detection and the accuracy of bTMB assessment. Additionally, biological factors like the tumour fraction in plasma cell-free DNA (cfDNA) and the heterogeneity within circulating tumour DNA (ctDNA) can impact the results [35].

Consequently, the agreement between bTMB and tissue-based TMB (tTMB) may be low, which can complicate the interpretation of the bTMB results and limit its effectiveness as a biomarker for predicting responses to ICIs [35]. Using paired testing of both tumour and blood-based TMB assessments may help mitigate variations caused by tumour heterogeneity [36]. The clinical application of blood-based TMB (bTMB) using ctDNA necessitates further standardization to establish it as a dynamic biomarker. This is especially important when monitoring cancers over time, particularly in situations where repeated tumour biopsies are difficult to obtain. Efforts to harmonize genomic profiling and predict responses to immune checkpoint inhibitors (ICIs) will need to focus on evaluating biomarkers from circulating tumour cells (CTCs), ctDNA, and extracellular vesicles (EVs) [35]. Secondly, unlike PD-L1, which can only predict the response to PD-1/PD-L1 inhibitors, TMB can forecast the response to various immunotherapies, including PD-1/PD-L1 inhibitors and anti-CTLA4 antibodies (such as ipilimumab) [37].

In addition to PD-L1, TMB can also be correlated with other biomarkers, such as GEP, to predict patients’ responses to immunotherapy. In this context, the study led by Cristescu et al. [38] showed that TMB and GEP had a modest correlation with each other, and that both were independently able to predict patient responses across the KEYNOTE clinical trials. Their analysis revealed that ORRs were higher in patients with both high GEP and high TMB (37% to 57%), moderate in those with high GEP and low TMB (12% to 35%) or low GEP and high TMB (11% to 42%). The lowest response rate or no response were observed in patients with both low GEP and low TMB (0% to 9%). Furthermore, patients with elevated levels of both TMB and GEP had longer PFSs. The results were consistent when evaluating TMB alongside PD-L1 expression. This analysis demonstrates that TMB, along with inflammatory biomarkers such as T cell-inflamed GEP and PD-L1 expression, can categorize human cancers into distinct groups with varying clinical responses to immunotherapy. It also helps to identify the underlying, targetable biological patterns associated with these groups. Both TMB and inflammatory biomarkers independently predict therapeutic responses, potentially reflecting different aspects of neo-antigenicity and T cell activation. This method could offer a framework for future precision medicine, aiding in the rational design and assessment of combination therapies involving anti–PD-1 and/or anti–PD-L1 treatments [37].

The study focused on highlighting the response to pembrolizumab monotherapy based on TMB and GEP. Therefore, patient groups categorized by TMB and GEP status exhibit significant variations in their clinical responses to Pembrolizumab. Specifically, the groups with only one positive biomarker—either high TMB and low GEP or low TMB and high GEP—demonstrate considerably lower response rates compared to the group with both high TMB and high GEP. This suggests the possible presence of resistance mechanisms to Pembrolizumab. To identify potential resistance mechanisms, several molecular analyses were conducted [37].

The inclusion of biomarkers such as PD-L1 expression and gene expression profiling (GEP) alongside tumour mutational burden (TMB) can address the limitations of TMB as a standalone predictor, particularly in cancers like breast and prostate. These tumour types often exhibit lower mutational burdens, making the integration of additional biomarkers critical for improving predictive accuracy.

PD-L1 reflects tumour-immune interactions, while GEP provides insights into immune-related gene activity, offering complementary perspectives on the genomic data captured by TMB.

To advance this field, research should focus on developing integrated diagnostic frameworks that combine TMB, PD-L1, and GEP into unified panels. Such tools could streamline workflows, improve patient stratification, and enable more personalized treatment strategies across diverse cancer types.

The incidence of somatic mutations varies across different tumour types, with NSCLC exhibiting the highest mutation frequency, ranging from 0.1 to 100 mut/Mb [35]. A retrospective analysis performed by Rizvi et al. [39] demonstrated that the efficacy of ICI therapy in NSCLC patients is related to the TMB value. Patients with high TMB demonstrated better efficacy and higher survival rates compared to those with a lower TMB. Klempner [40] additionally observed that a TMB cut-off value of 10 mut/Mb could predict the efficacy of ICIs in NSCLC patients, with higher TMB thresholds correlating with longer progression-free survival. In most of the NSCLC studies, targeted Next Generation Sequencing (NGS) suggests a TMB cut-off value of around 10 mut/Mb. While many studies have shown the predictive value of TMB for ICI therapy in NSCLC patients, some have reported negative outcomes, particularly concerning long-term survival. This discrepancy may be due to the limited attention and research on TMB in this context [36].

In addition to the studies related to lung cancer, the specialized literature also includes studies analyzing cases of malignant melanoma treated with ICIs. In Eckardt’s study [41], the effectiveness of using TMB as a biomarker for patients treated with ICIs was demonstrated. Furthermore, the study’s approach to the BRAF mutation, identified as a potential predictive biomarker for cases treated with targeted therapies, such as Dabrafenib, was also noteworthy. Those two markers were classified as independent viable predictive biomarkers for a relapse-free survival period (ranges between 21 and 100%) in patients diagnosed with malignant melanoma. Melanomas with BRAF mutations associated with high TMB are likely to benefit from adjuvant anti-PD-1 therapy. Conversely, patients with low TMB may gain more from adjuvant BRAF and MEK inhibitors, assuming that their tumour tissue is less heterogeneous. However, the lack of direct comparative studies on this topic means that both adjuvant treatment options should be discussed with patients who have BRAF mutations [40].

Lastly, prostate and breast malignancies have a poor response to ICI treatment, but in recent years, important improvements have been made in this direction. Because of that, several studies have started to investigate the importance of the TMB value in these types of cancers. There is one preliminary study [42] that analyzed 48 patients diagnosed with metastatic prostate/breast cancer and demonstrated that TMB, in these cases, was not directly associated with a better treatment response to ICIs. Instead, it was associated with a significantly higher number of genomic alterations and more pronounced MSI. These findings suggest that, with further research, TMB and MSI could potentially be correlated with a favourable response to ICIs in the future. Although minimal activity was observed in the group of patients with blood TMB evaluation/MSI not detected, clinical benefit was noted in patients with notable MSI defects [42].

Our meta-analysis has some limitations, based on the presented information:Variation in TMB cut-off values: Some studies define high TMB ≥ 10 muts/Mb, while others may use other thresholds. This inconsistency in defining high TMB can lead to variability in results and limits the generalizability of the findings across different settings and patient populations;Inconsistencies in methodology: The measurement of TMB varies between studies, particularly in the use of different sequencing technologies. Some studies used NGS and others used WES, which could affect the sensitivity and accuracy of TMB assessment;Studied population: Many studies were retrospective, and the patient populations were heterogeneous, including different cancer types and treatment regimens. These variations can introduce bias and limit the ability to make generalized conclusions about TMB as a predictive biomarker across all cancers and immunotherapy treatments. Additionally, the small sample size in some studies may affect the robustness of the findings;Publication bias: Like many systematic reviews and meta-analyses, there is a risk of publication bias, where studies with positive results are more likely to be published, while studies with null or negative results may remain unpublished. This can lead to an overestimation of the effectiveness of TMB as a predictive biomarker for ICI effectiveness;Exclusion of non-English studies: If the review excluded non-English language studies, this could introduce language bias. Important studies conducted outside English-speaking countries might have been overlooked, potentially limiting the comprehensiveness of the review.

Although formal methods such as the funnel plot or Egger’s test method were not used to assess reporting bias, we observed that most of the studies included in the meta-analysis reported positive results. This suggests the possibility of reporting bias, where studies with negative or neutral results might be underreported.

## 8. Conclusions

As a conclusion, in our comprehensive analysis, TMB holds significant promise as a predictive biomarker for the response to immune checkpoint inhibitors across various cancer types. Our analysis of the current literature highlights a more established consensus regarding the importance of TMB in NSCLC and melanoma, where high TMB has been associated with better clinical outcomes. However, the role of TMB in breast and prostate cancers remains less clear, indicating a need for further research in these areas.

Despite the variability in findings, the potential of TMB to guide tailored immunotherapy treatment is undeniable. As we continue to explore and refine our understanding of TMB, especially in less well-defined cancers, it becomes essential to conduct large-scale, randomized studies. These studies will aid in standardizing TMB evaluation and confirming its predictive efficacy, ultimately advancing the precision of cancer treatment and enhancing patient prognoses.

In addition to PD-L1 and TMB, there are several other parameters associated with immune checkpoint inhibitor outcomes, such as prognostic scores like LIPI and dNLR, tumour-infiltrating lymphocytes (TILs), T-effector/IFN-γ signatures, general immune fitness, soluble inhibitors, tumour metabolism, and the microbiome. Combining these parameters could provide valuable insights. However, a challenge with integrating multiple biomarkers is the increasing number of potential patient subgroups. This complexity makes it harder to establish cut-offs and understand interdependencies. Most clinical trial reports have only combined PD-L1 and TMB. Additionally, obtaining sufficient tissue for comprehensive analyses—such as defining histological subtypes, testing molecular drivers, and evaluating PD-L1 and TMB—can be challenging. Moreover, many predictive biomarker tests are costly for routine use. Exceptions include predictive scores based on routinely collected laboratory values, such as the LIPI score, which uses only pretreatment neutrophil, lymphocyte, and LDH levels. The performance of these scores should be assessed and compared with PD-L1 and TMB testing [43].

## Figures and Tables

**Figure 1 cancers-17-00480-f001:**
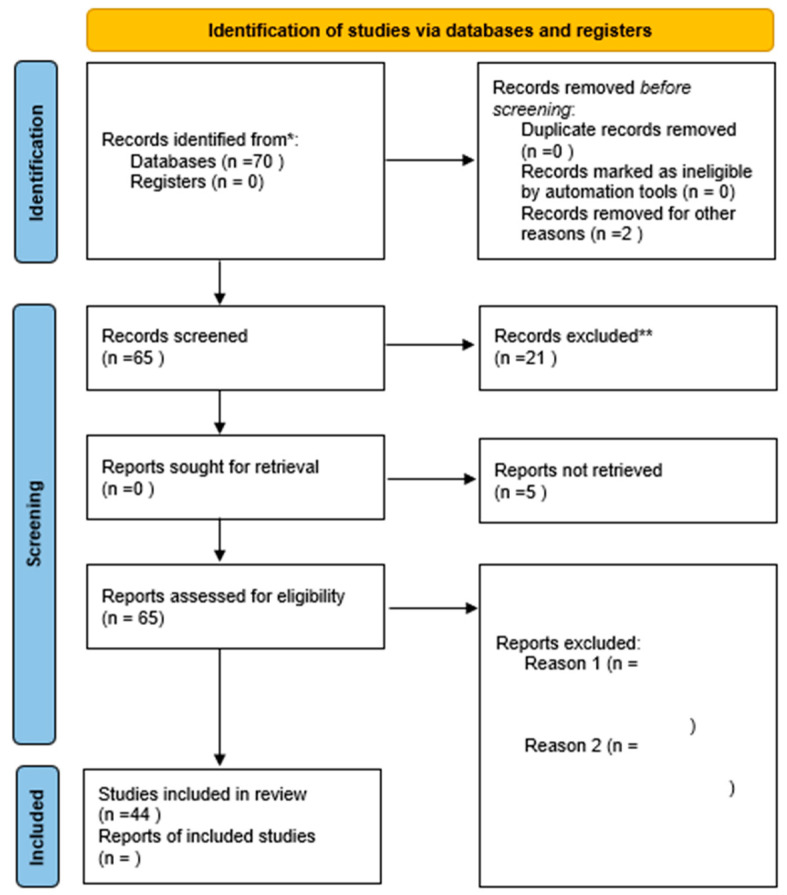
PRISMA flow chart. * Consider, if feasible to do so, reporting the number of records identified from each database or register searched (rather than the total number across all databases/registers); ** If automation tools were used, indicate how many records were excluded by a human and how many were excluded by automation tools.

**Table 1 cancers-17-00480-t001:** Comparison between the three main studies regarding TMB use in NSCLS.

Study	Studied Treatment	Primary Endpoint/s	Secondary Endpoint/s
CheckMate 227	Nivolumab + Ipilimumab vs. Platinum-based chemotherapy	PFS based on TMB OS based on PD-L1	PFS based on TMB OS based on TMB
CheckMate 568	Nivolumab + low dose Ipilimumab	ORR based on PD-L1	ORR/PFS/OS/efficacy by TMB and PD-L1
CheckMate 026	Nivolumab vs. Platinum-based chemotherapy	PFS based on PD-L1	PFS/OS/ORR based on PD-L1

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
