# Peer review of "Evaluating Tumour Mutational Burden as a Key Biomarker in Personalized Cancer Immunotherapy: A Pan-Cancer Systematic Review"

_cancers, 2025, doi:10.3390/cancers17030480_

Round 1

Reviewer 1 Report

Comments and Suggestions for Authors

The paper contextualizes the significance of TMB in cancer immunotherapy, emphasizing its prominence as a biomarker for predicting the efficacy of ICIs.

My concerns:

 It suggests the predictive utility of high TMB in lung cancer and melanoma but highlights the inconclusive results for breast and prostate cancers.

The identification of methodological inconsistencies, such as variations in TMB assessment techniques, is a critical observation that underscores the need for standardized protocols.

it could benefit from a more in-depth exploration of the potential reasons behind the variability in TMB's predictive value across different cancer types.

the variation in TMB cut-off values, inconsistencies in methodology, and potential publication bias.

It calls for large-scale, randomized studies to further validate TMB's predictive efficacy and standardize its evaluation.  

Major:

  1. More detail prioritize the development of standardized protocols for TMB assessment to address the methodological inconsistencies identified in the review.
  2. Investigating complementary biomarkers and their integration with TMB could enhance the predictive accuracy for breast and prostate cancers.

The review's findings underscore the need for standardized TMB assessment protocols and further research to refine its use in personalized cancer therapy.

Author Response

1.More detail prioritize the development of standardized protocols for TMB assessment to address the methodological inconsistencies identified in the review.

Evaluating TMB is crucial in personalizing oncological treatments, particularly in immunotherapies. NGS methods are fundamental for determining the number of somatic mutations in tumour DNA. These methods differ significantly depending on the targeted genomic region:

  1. Whole Genome Sequencing (WGS) provides a comprehensive analysis, including both coding and non-coding regions of the genome.
  2. Whole Exome Sequencing (WES) focuses on coding regions, which represent only 1-2% of the genome but contain most cancer-related mutations.

Recently, the FDA approved FoundationOne CDx and MSK-IMPACT, 2 gene panel assays designed to identify genetic alterations in solid tumours. These assays are used in clinical settings due to their precision and ability to streamline TMB assessment. However, the diversity of platforms and methodologies used in different studies has led to substantial variability in TMB results, complicating the comparison of findings and their clinical applicability.

This highlights the pressing need for standardized protocols at an international level. Standardization would ensure consistency in the methods used, including sequencing platforms, bioinformatics pipelines, and TMB cutoffs. Collaborative efforts between regulatory agencies, academic institutions, and industry stakeholders are essential to establish these standardized frameworks. Furthermore, the integration of validated assays like FoundationOne CDx into unified protocols could serve as a model for harmonizing TMB evaluation in clinical and research settings.

2. Investigating complementary biomarkers and their integration with TMB could enhance the predictive accuracy for breast and prostate cancers.

The inclusion of biomarkers such as PD-L1 expression and gene expression profiling (GEP) alongside tumour mutational burden (TMB) can address the limitations of TMB as a standalone predictor, particularly in cancers like breast and prostate. These tumour types often exhibit lower mutational burdens, making the integration of additional biomarkers critical for improving predictive accuracy.

PD-L1 reflects tumor-immune interactions, while GEP provides insights into immune-related gene activity, offering complementary perspectives to the genomic data captured by TMB.

To advance this field, research should focus on developing integrated diagnostic frameworks that combine TMB, PD-L1, and GEP into unified panels. Such tools could streamline workflows, improve patient stratification, and enable more personalized treatment strategies across diverse cancer types.

Reviewer 2 Report

Comments and Suggestions for Authors

The article used a simple format organization, which is easy to follow. Specific comments are below.

1. Try to be consistent for everything throughout the manuscript. For example some abbreviations were bolded while other not.

2. This reviewer does not suggest bolding both full name and abreviations such as the authors did for hazard ratios (HRs)  ….. Others such as Review Manager (RevMan 5.4). the Review Manager may not need to be bolded.

3. The references should be more efficiently cited in the entire manuscript to let readers know where the information coming from. For example, the while two paragraphs of description below did not cite a reference. “In this study, a total of men were evaluated between January 2011 and April 2021. They were subjected to genomic testing through the comprehensive genomic profiling 368 (CGP) assays provided by Foundation Medicine. Patients were included in this study if they received either single-agent anti–PD-1 axis therapy or single-agent taxane in the 370 mCRPC setting and had their TMB assessed through tissue biopsy. The main clinopatho-371 logical characteristics were represented by the median age, 70 (ranges between 64-76 372 years) and the baseline median pretreatment PSA levels of 79.4 ng/ml. A total of 108 pa-373 tients (18.8%) had ECOG scores of 2 or greater, and 644 patients (86.9%) had received prior 374 systemic treatments for mCRPC. A total of 45 patients (6.1%) received ICIs, while 696 pa-375 tients (93.9%) received taxanes. Patients who received ICIs and those who received tax-376 anes showed no significant differences in age, pretherapy PSA levels, ECOG scores, prior NHT use, prior prescribed opioid use, and biopsy site. However, it is important to note 378 that patients who received ICIs had higher TMB compared to those receiving taxanes (3.5 379 mut/Mb vs. 2.5 mut/Mb; P < .001).” “PSA levels were evaluated in 607 patients. Among them, 14 patients had a TMB above 381 10 mut/Mb and were treated with ICIs. Of these, 4 exhibited a reduction in PSA levels by 382 approximately 50%. In contrast, none of the patients with a TMB below 10 mut/Mb 383 showed a reduction in PSA levels greater than 50%. For patients treated with taxanes, no 384 relationship between TMB value and PSA level was demonstrated.

4. After a long Discussion, the Conclusion section should be as short as possible for readers to get a clearer conclusion in mind.

Author Response

1.Try to be consistent for everything throughout the manuscript. For example some abbreviations were bolded while other not... I have reviewed the manuscript and adjusted the bolded abbreviations. The changes made are marked in red.

2.This reviewer does not suggest bolding both full name and abreviations such as the authors did for hazard ratios (HRs)  ….. Others such as Review Manager (RevMan 5.4). the Review Manager may not need to be bolded....I made some modifications; are marked in red.

3. The references should be more efficiently cited in the entire manuscript to let readers know where the information coming from...I made the modifications for those specific paragraphs (lines 375 & 393- marked in red)—additional changes: lines 272, 333.

4. After a long Discussion, the Conclusion section should be as short as possible for readers to get a clearer conclusion in mind....

In conclusion, our analysis underscores TMB as a promising predictive biomarker for immune checkpoint inhibitors across various cancers. While its role is well-established in NSCLC and melanoma, further research is needed to clarify its utility in breast and prostate cancers. Large-scale studies are crucial to standardize TMB evaluation and confirm its predictive efficacy, paving the way for more precise cancer treatments.

Additionally, integrating TMB with other biomarkers, such as PD-L1, immune scores, and tumor-infiltrating lymphocytes, could provide deeper insights. However, challenges like subgroup complexity, tissue availability, and high testing costs must be addressed to enhance their clinical applicability.

Round 2

Reviewer 1 Report

Comments and Suggestions for Authors

accepted on current form